# Revealing internal flow behaviour in arc welding and additive manufacturing of metals

Lee Aucott [1], Hongbiao Dong [2], Wajira Mirihanage[3], Robert Atwood [4], Anton Kidess[5,10], Shian Gao[2], Shuwen Wen[6,11], John Marsden[6], Shuo Feng[2], Mingming Tong[7,12], Thomas Connolley [4], Michael Drakopoulos[4], Chris R. Kleijn[5], Ian M. Richardson[8], David J. Browne[7], Ragnvald H. Mathiesen[9] & Helen.V. Atkinson[2,13]

Internal flow behaviour during melt-pool-based metal manufacturing remains unclear and hinders progression to process optimisation. In this contribution, we present direct time-resolved imaging of melt pool flow dynamics from a high-energy synchrotron radiation experiment. We track internal flow streams during arc welding of steel and measure instantaneous flow velocities ranging from $0.1\,\mathrm{m\,s^{-1}}$ to $0.5\,\mathrm{m\,s^{-1}}$. When the temperature-dependent surface tension coefficient is negative, bulk turbulence is the main flow mechanism and the critical velocity for surface turbulence is below the limits identified in previous theoretical studies. When the alloy exhibits a positive temperature-dependent surface tension coefficient, surface turbulence occurs and derisory oxides can be entrapped within the subsequent solid as result of higher flow velocities. The widely used arc welding and the emerging arc additive manufacturing routes can be optimised by controlling internal melt flow through adjusting surface active elements.

[1] United Kingdom Atomic Energy Authority, Culham Science Centre, Abingdon, Oxfordshire OX14 3DB, UK. [2] Department of Engineering, University of Leicester, Leicester LE1 7RH, UK. [3] School of Materials, University of Manchester, Manchester M13 9PL, UK. [4] Diamond Light Source, Didcot OX11 0DE, UK. [5] Department of Chemical Engineering, Delft University of Technology, 2629HZ Delft, The Netherlands. [6] Tata Steel, Research & Development, Swindon Technology Centre, Rotherham S60 3AR, UK. [7] School of Mechanical and Materials Engineering, University College Dublin, BelfieldDublin 4, Ireland. [8] Department of Materials Science and Engineering, Delft University of Technology, 2629HZ Delft, The Netherlands. [9] The Norwegian University of Science and Technology, Høgskoleringen 1, 7491 Trondheim, Norway. [10] Present address: Hilti Entwicklungsgesellschaft mbH, Computational Engineering, Hiltistraße 6, 86916 Kaufering, Germany. [11] Present address: Dongguan Centre of Excellence for Advanced Materials, Dongguan 523808 Guangdong, China. [12] Present address: National University of Ireland Galway, Galway H91 TK33, Ireland. [13] Present address: School of Aerospace, Transport and Manufacturing, Cranfield University, Cranfield MK43 0AL, UK. Correspondence and requests for materials should be addressed to H.D. (email: h.dong@le.ac.uk)

During fusion welding or additive manufacturing of metals, a localised heat input induces rapid and successive phase transformations from solid-to-liquid and then liquid-to-solid through the formation of a molten metal pool. The formation and control of the melt pool is one of the crucial elements of fusion-based advanced manufacturing processes. Melt pools formed during fusion welding solidify to form bonded joints in high integrity products such as automobiles, ships, transcontinental oil pipelines, and most other large metallic structures. Additive manufacturing techniques are similarly based on the concept of successive creation of diminutive melt pools. Correct control of the melt pool is essential in these advanced manufacturing processes to avoid catastrophic failures which could lead to considerable humanitarian, economic and environmental damage. In the manufacturing process, electric arcs, flames, plasma, lasers or electron beams can be selected as a heating source according to the manufacturing method. On cooling, the molten metal pool solidifies with a different microstructure to that of the original material. In welding, the newly solidified material is commonly termed the fusion zone. In the case of additive manufacturing, the whole component can be considered as a collection of successive fusion zones or deposits. Flow dynamics and geometrical evolution of the liquid melt pool have been suggested to have strong relationships with the subsequent mechanical properties of additively manufactured[1–3] and welded materials[4–6]. Consequently, the lifetime and performance of the additively manufactured components or welded joints, and hence the entire structural or functional components, can be significantly dictated by the geometric evolution and flow dynamics of the melt pool[1,2,7–10]. Several fundamental physical phenomena have been identified as governing factors for melt pool fluid flow; these include buoyancy/convention, electromagnetic (Lorentz) force, plasma/arc drag, and surface tension effects[11,12]. The prevailing forces may be characteristic of the heat source being employed, or of the materials being processed. For steels, it has been suggested that surface tension-driven Marangoni forces dominate the melt pool flow[13], and is a key to determining the shape and penetration of the solidified joint. It may also have an influence on the final residual stresses, and can set the conditions for the origin of defects in welded joints[14–25] or additively formed components[2,3].

In order for melt pool flow and evolution to be better understood, significant efforts[2,13,18,25–32] have been placed upon computational modelling of fluid flow and heat transfer in welding. Typically, model predictions were compared to final geometrical and structural features, without attention to the transient nature of the process[7]. Some validation of models has been attempted with data from simulated transparent systems such as $NaNO_3$[33–35]. Such experiments, studying transparent systems, can only roughly guide computational models, as distinct physical property-driven differences exist between these analogous materials and metallic alloys; for example, in the Prandtl number. Such distinctive dissimilarities make metal analogous materials an insufficient tool for the rigorous validation of melt pool models. Thus, there is a need for real-time melt pool evolution data in welding or additive manufacturing, generated with real metallic alloys, such as steels or light metal alloys. Provision of such results would solve one of the key obstacles thwarting the progress of advanced computational models of welding, e.g. refs. [9,10,18,26,36–38]. Common flow quantification techniques, such as positron emission particle tracking[39], particle image velocimetry, or laser Doppler anemometry[40], cannot be applied due to the high temperature, opacity, and fast dynamics of the liquid flow in metallic melt pools. However, application of very bright high-energy X-rays, generated from synchrotron sources, has opened interesting new avenues for experimental exploration. For example, Zabler et al.[41] used an in situ micro-radiography technique to observe particle and liquid motion in semi-solid Al alloys. Mirihanage et al.[42], implemented fast synchrotron X-ray diffraction to study rapid liquid–solid phase transformations in welding. Aucott et al.[43,44] employed fast synchrotron X-ray radiography to investigate initiation and growth kinetics of solidification cracks within a melt pool during welding. Leung et al.[45] used a similar technique to study defect and molten pool behaviour in powder based laser additive manufacturing, where melt pools are relatively small and melt vaporisation and recoil pressure play a dominant role compared to the case presented in our contribution here.

In this study, a high-energy synchrotron micro-radiography technique is employed to observe melt pool formation and flow dynamics during advanced manufacturing of metallic alloys. Time-resolved images are used to quantify morphological evolution and flow dynamics within the melt pool in situ. The widely used arc welding and the emerging arc additive manufacturing routes can be optimised by controlling internal melt flow through adjusting surface active elements.

## Results

**In situ X-radiography.** In situ experiments were performed on the I12 (JEEP) beamline at Diamond Light Source, UK[46]. Melt pools were created in solid steel bars using an electric arc generated from Tungsten Inert Gas (TIG) welding equipment. The experimental setup, illustrated in Fig. 1, was positioned to transmit the incoming high-flux synchrotron white beam through the molten metal pool. X-ray radiographs of the molten region, illustrated in the bottom image of Fig. 1, were captured by employing a scintillator coupled fast CMOS camera, at 1 or 2 kHz frame rates, covering the whole molten region in the field of view (FoV).

Tungsten (W) and tantalum (Ta) particles, ~50 μm in size, were employed as tracers to visualise the flow in the melt pool by investigating their spatiotemporal distribution. Due to their high melting points, the particles remain solid in the melt pool for a sufficient amount of time for them to be tracked. The particles were placed on top of the sample surface before the start of the rapid melting process. As the melt pool began to form, the particles were immersed into the molten metal and moved according to the flow. In comparison to the iron and other constituting elements of the sample material, W and Ta particles exhibit significantly higher X-ray attenuation. Thus, those solid particles in the melt pool appear darker than the surrounding liquid metal in the projected images, as illustrated in the example radiograph in Fig. 1. While the particles will likely affect the microstructure of the solidified fusion zone, they have an insignificant effect on the flow patterns and velocities in the liquid melt pool as the sinking speed is negligible in comparison to the highly dynamic flow velocities characterised.

**Morphological evolution of melt pool.** It is known that low S and high S steels, when welded under exactly the same conditions, exhibit post solidification weld shapes that differ distinctively[47]. The morphological (geometric shape) evolution of low S (0.0005 wt% S) and high S (0.3 wt% S) steel melt pools are presented in Fig. 2. Both melt pools were created using exactly the same process parameters and sample dimensions. The images represent three instances to assess the overall geometric evolution in respect to time, until the melt pools grow to their maximum size. The graphs within Fig. 2 quantify melt pool evolution in terms of width, depth and width/depth ratio at 100 ms intervals. The maximum temporal resolution available is 1 ms, but 100 ms is sufficient to quantify geometric shape evolution. Qualitative

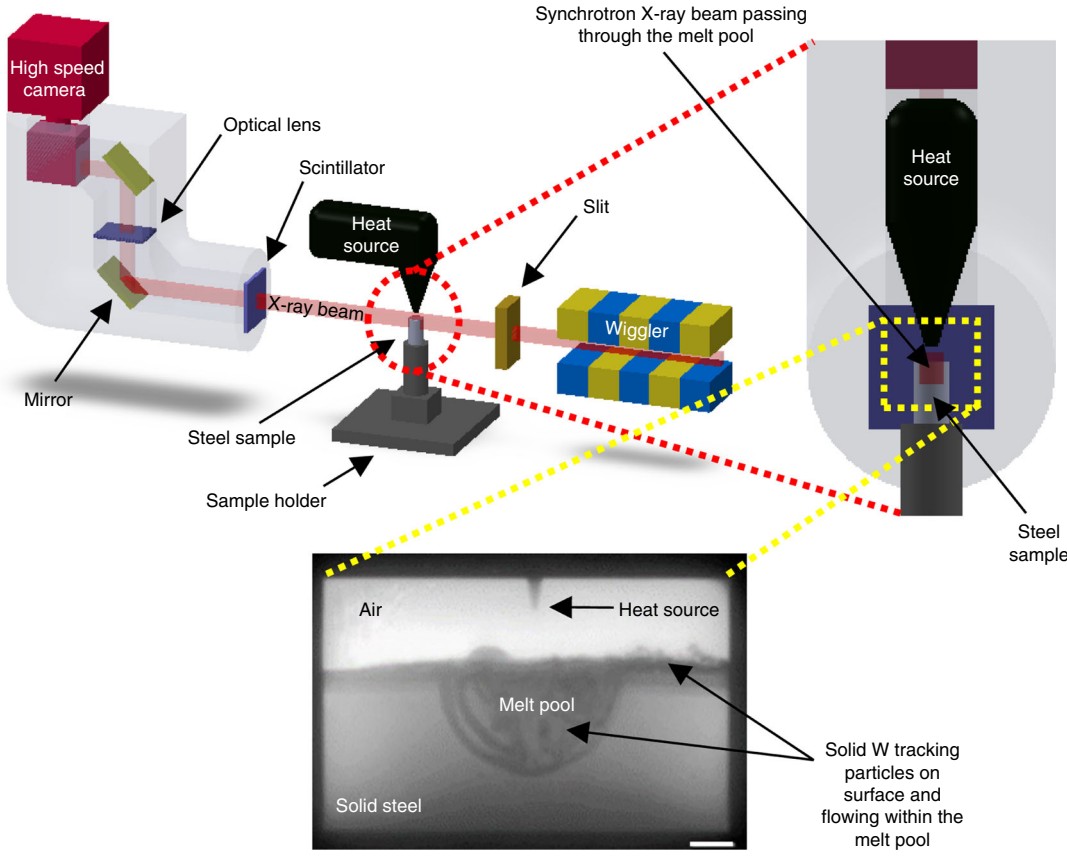

**Fig. 1** Schematic diagram of the experimental setup and an example radiograph annotated to show the key elements under observation during the experiment. A polychromatic (white) beam of ~50–150 keV was used to maximise the X-ray photon flux. The beam size was $12 \times 50$ mm$^2$ ($H \times W$) and was transmitted through the entire melt pool. The detector was a Vision Research Phantom v7.3 CMOS camera, lens-coupled to cadmium tungstate or cesium iodide scintillators. With an optical magnification of ×1.8, the linear resolution was 13 μm per pixel. Imaging was acquired at frame rates up to 2 kHz at $800 \times 600$ pixels per frame. Scale bar $= 1$ mm

analysis of the radiographs shows that the evolution of the melt pool shape significantly differs between the two samples. Quantitative measurements of melt pool width evolution show that in the initial 500 ms the width evolution is nearly identical. After 500 ms, the low S sample begins to grow wider than the high S sample. The difference in the depth evolution is much more pronounced between the two samples. The high S sample immediately grows at a higher rate than the low S sample and continues to grow further from that of the low S sample as the melt pool evolves. The melt pool of the low S sample reached only 1.34 mm depth after 2000 ms from the start of the melting. The high S steel melt pool appears to favour downward growth to penetrate a depth of 3.59 mm, over 150% deeper than the low S melt pool. As a result, the aspect (width to depth) ratio of the melt pool in the low S sample is three times higher than the high S sample throughout melting. The aspect ratio is also very consistent throughout the melting process, especially in the high S sample. Furthermore, the total volume of alloy melted after 2000 ms is ~200% greater in the high S (~72.47 mm$^3$) sample compared to the low S (~23.84 mm$^3$) sample, yet the heat input to both alloys is the same.

**Flow dynamics**. Melt pool flow dynamics have been quantified to rationalise the morphological evolutions observed in Fig. 2. The maximum temporal resolution of the experiment (1 ms) was exploited to capture the fast flow dynamics in the melt pool. For low S steel, the flow dynamics are illustrated in Fig. 3. The

coloured lines indicate the travel path of tracer particles. The tracer particles were tracked over an 80 ms time span (across 80 consecutive frames), starting at ~1 and ~2 s, respectively, from the inception of melting, after igniting the arc. Flow pattern observations suggest that melt pool shape evolution is mainly determined by the characteristics of flow. At both time instances (~1 s and ~2 s) illustrated in Fig. 3, the tracer particles follow anticlockwise paths in the left-half of the melt pool and clockwise paths in the right-half of the melt pool cross section, i.e. there is an outward flow in the upper part of the melt pool, and in inward flow in its lower part. Consequently, the highest temperature liquid metal under the heat source is being convectively transported horizontally away from the melt pool centre towards its lateral extremities. This stimulates growth of the shallow and wide melt pool depicted in Figs. 2 and 3.

The experimental observations for high S steel melt pools indicate flow in the opposite direction to the low S steel, as illustrated through a representative example in Fig. 4. The flow patterns tracked by tracer particles reveal inward flow in the upper part of the melt pool, and outward flow in its lower part with a close symmetry of the projected melt pool cross section. It appears, in high S steel, the highest temperature liquid metal in the upper centre region of the melt pool is transported vertically downwards to the centre bottom of the melt pool. Thus, the bottom of the melt pool receives more heat load, which stimulates further melting of the solid substrate beneath the solid–liquid interface at the bottom of the melt pool. As a result, the melt pool depth/width ratio increases.

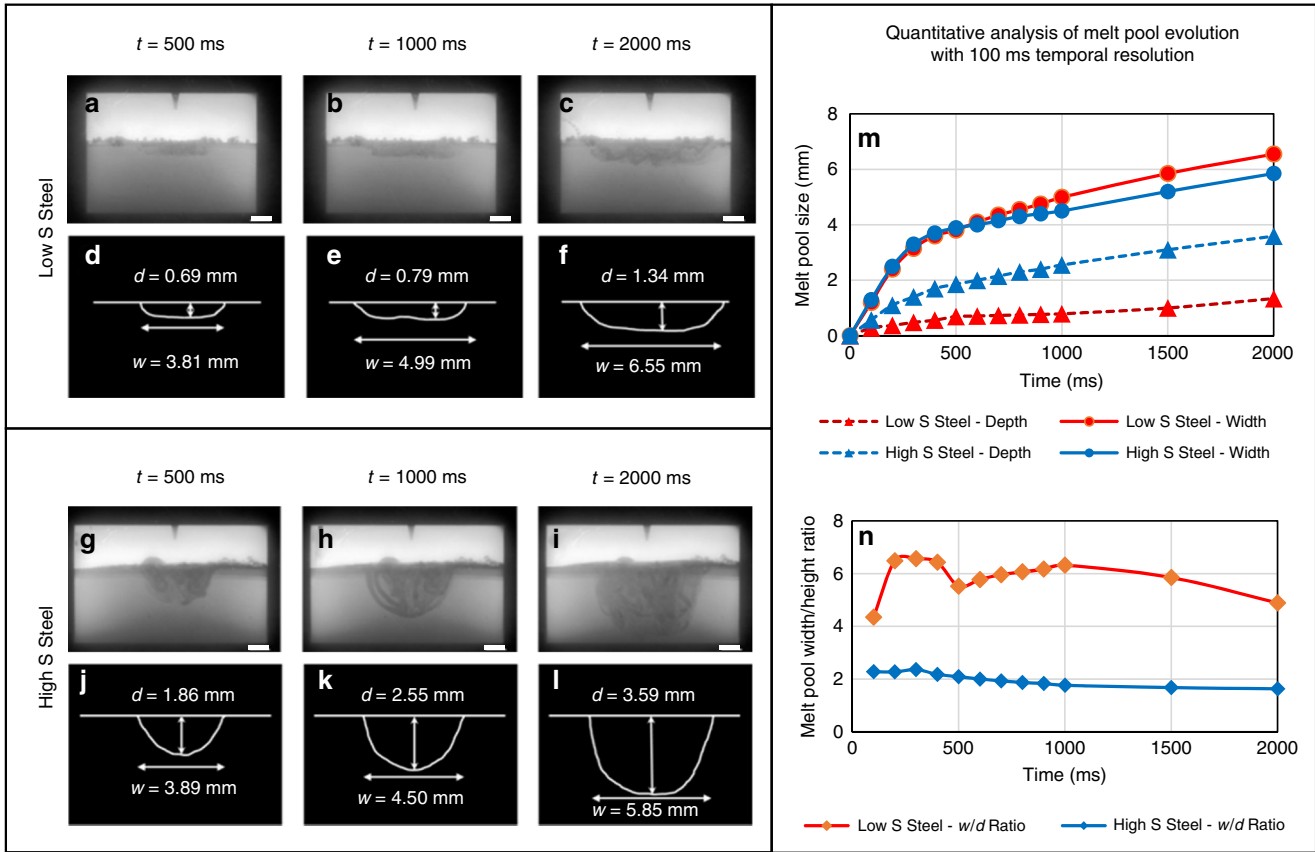

**Fig. 2** Quantitative analysis of time dependent evolution of melt pool morphology. **a–l** Synchrotron X-ray radiographs of the evolving melt pool at three-time instances in situ. The corresponding measured geometries are below the respective radiographs. The melt pools are created using the same melting parameters and sample dimensions. Panels **a–f** are from a low S steel melt pool, while **g–l** are from a high S steel melt pool. Melt pool size evolution is quantified with 100 ms temporal resolution in **m** and **n**. It is evident that the high S melt pool favours downward growth to penetrate a depth of 3.59 mm, over 150% deeper than the low S melt pool. All scale bars = 1 mm

An indication of approximate instantaneous flow velocities can be determined from the measurement of distance and time between consecutive tracer positions. Figure 5 shows velocities for low and high S steel melt pools. These figures suggest that the flow velocities can exceed $0.5\,\mathrm{m\,s^{-1}}$ in high S steels, whereas, in low S steels, flow velocities did not exceed $0.3\,\mathrm{m\,s^{-1}}$.

## Discussion

The results presented in this study demonstrate the capability of synchrotron X-ray imaging to capture, visualise and particularly quantify melt pool evolution in situ in real metallic alloys for the first time. These novel experimental observations allow us to observe and comprehend how melt pool formation and evolution progress under realistic fusion welding conditions or other relatively moderate size melt pools associated with manufacturing processes such as wire arc additive manufacturing.

Our observations confirm some theoretically and computationally predicted melt pool forming mechanisms. For example, they agree qualitatively with high speed motion pictures of melt pool size evolution[30,48] and confirm the suggested mechanisms for formation of fusion zone shape (shallow with shrunken centres for low S steels, deep with risen centres for high S steels) through metallography analysis of weld joints[49,50]. Changing the steel composition from low S to high S not only changes the flow direction, but also increases the flow velocities and melt pool dimensions. As a result, Reynolds numbers in high S weld pools are up to 2.5 times larger than in low S weld pools. Furthermore,

the total volume of alloy melted is ~200% greater in the high S sample yet the heat input to both alloys is the same. As a result, one would assume a lower superheat in the liquid of high S steel and/or a lower temperature gradient $G$. This would in turn lead to a different evolving Hunt G–V diagram[51] in both cases, and therefore, a different columnar to equiaxed transition (CET) and equiaxed zones. This behaviour was predicted by Kidess et al.[36] and proves a strong evidence of the link between surface energy/temperature derivative, weld pool shape and microstructure.

Related to metal casting, Campbell[52] classified two types of flow within solidifying metals. Firstly, bulk turbulence, i.e. the chaotic eddying flow of the bulk liquid, assessed by Reynolds number, and secondly, surface turbulence, which causes the chaotic breaking up of the surface of the liquid, allowing the surface oxide film to become incorporated into the bulk melt. While the avoidance of bulk turbulence is probably impossible in welding, Campbell's derived estimation for the critical velocity for surface turbulence in liquid metals is $\sim\!0.5\,\mathrm{m\,s^{-1}}$. The flow velocities quantified in this study must be assessed in relation to the melt pool in which they exist, the measured velocity is lower than those predicted value of $1\,\mathrm{m\,s^{-1}}$ by modelling in very large size melt pool. In this study, the 10 mm sample was specifically chosen as it was the maximum size feasible for the X-ray beam to penetrate and provide high resolution images for subsequent analysis. Other factors may also affect the discrepancy between the measured velocity and the simulated ones. Firstly, we are essentially measuring the speed in a 2D projection, which may be much lower than the true speed in three dimensions, as we will

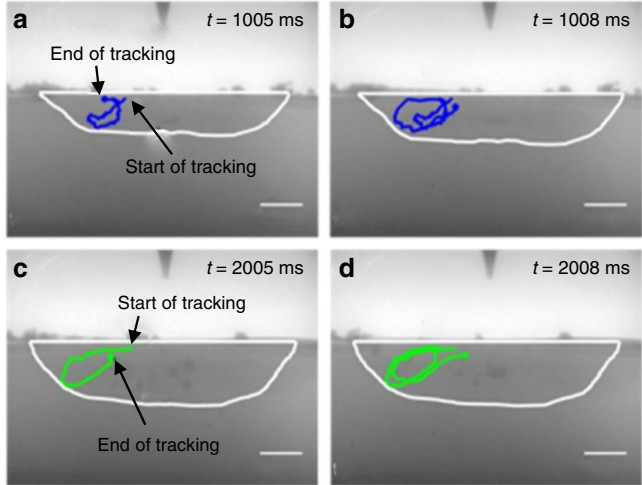

**Fig. 3** Fluid flow observed in low S steel melt pool. Tracer particles are tracked using a 1 ms temporal resolution—the maximum temporal resolution available. **a** Tracks a tracer particles movement from 1.00 s to 1.005 s. The loci joining each tracer position show the path of the particle and indicates an anti-clockwise flow path in the left-hand side of the melt pool with outward flow in the upper part of the melt pool, inward flow in the lower part, and upward flow along the centre of the melt pool. **b** Repeated recirculation of the particle tracked from 1.00 to 1.008 s. Two-time instances are assessed to demonstrate consistency throughout the melt pools life and **c** Identical tracer particle flow orientation tracked from 2.00 to 2.005 s. **d** Repeated recirculation of the particles tracked from 2.00 to 2.008 s. All scale bars = 1 mm

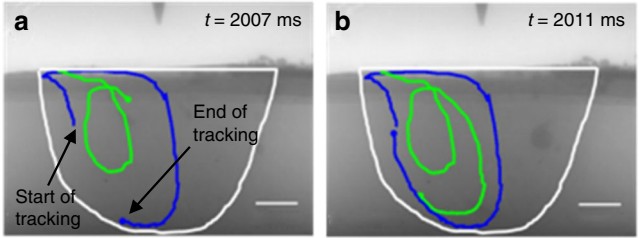

**Fig. 4** Fluid flow observed in high S steel melt pool. Tracer particles are tracked using a 1 ms temporal resolution—the maximum temporal resolution available. **a** Tracks two tracer particles movement from 2.00 to 2.007 s. The loci joining each tracer position show the path of each particle and indicates an opposite flow path to that observed in Fig. 3. In the high S steel melt pool, a clockwise flow path is observed in the left-hand side of the melt pool with inward flow in the upper part of the melt pool, downward flow along the centre, and outward flow in the lower part of the melt pool. **b** Repeated recirculation of the particles tracked from 2.00 to 2.011 s. All scale bars = 1 mm

likely never encounter a particle travelling exactly perpendicular to the imaging plane. Secondly, referring to computed results in literature may be misleading, since most papers neglect turbulence in their computations. However, turbulence will reduce the maximum velocity obtained due to increased thermal and momentum diffusivities. The quantified flow velocities from the experimental results presented in this study show that low S steels exhibit flow velocity values below this critical range for the given process parameters, as well as relatively low bulk Reynolds numbers. This suggests the absence of surface turbulence and probably also of bulk turbulence for such materials for the given operating conditions. Conversely, the higher flow velocities and higher bulk Reynolds numbers in the high S steel melt pool, highlight the potential for both bulk and surface turbulence and oxide entrapment within the fusion zone. These findings are in agreement with the work of Kidess[18,53], Xiao & Den Ouden[54] and Mills et al.[55] but opposite to those of Kou et al.[56]

One of the key achievements of this research is in situ experimental confirmation of the profound influence of the variation of surface tension ($\gamma$) with temperature on the melt pool flow. When $d\gamma/dT$ is negative, such as in iron or low S steel, higher temperature at the centre of the melt pool introduces a lower surface tension compared to the value at edge of the melt pool, resulting in a radially outward flow[57]. Observations reported in this contribution qualitatively confirm the predictions from model simulations of such melt pools[18,36,57,58], and confirm predicted behaviour in a melt pool without surface active agent. However, high S content within the melt pool is widely accepted as a means to increase the depth of the weld. It changes the temperature-dependent surface tension co-efficient of molten iron from negative to positive. Our study rationalises its effect through the observed reversal of the flow orientation in the melt pool.

Different forces have been suggested in literature to drive the melt pool flow in additive manufacturing or welding, including surface tension gradients, buoyancy, electromagnetic forces, arc and plasma pressure[7,57,59]. Despite the presence of this array of driving forces, our experimental evidence shows a distinct change in flow patterns as a result of varying the surface tension. This strongly suggests that surface tension effects are the dominant melt pool flow driving force for the materials and conditions examined here.

An important final remark is that the reported flow observations were 2D radiographic projections, even though the weld pool flow is 3D in nature. In order to quantify flow velocities, it was essential to select tracer particles that moved in the plane orthogonal to the synchrotron beam. To ensure selection of such orthogonal in-plane particles, only particles that flowed at the

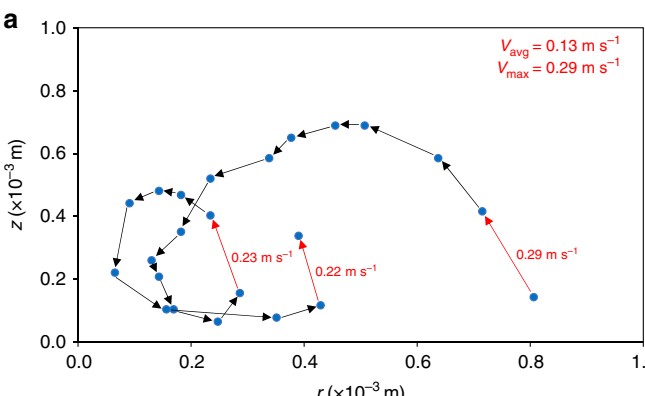

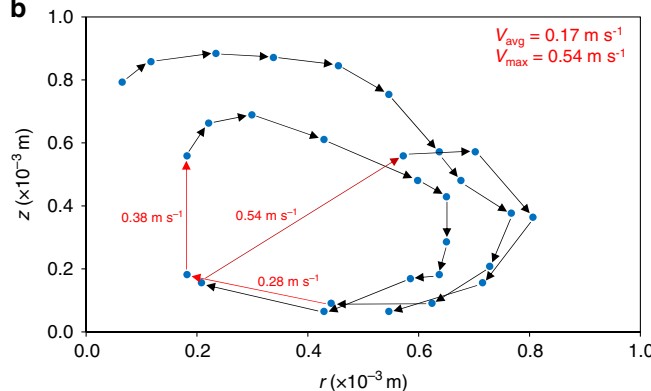

**Fig. 5** Instantaneous particle velocity measurements in **a** low S steel, and **b** high S steel ~1 and 2 s after the inception of melting, respectively. $r$ and $z$ denote the distance in the radial and vertical axis, respectively

extremities of the melt pool were tracked for quantifications. Existing X-ray tomographic methods to provide 3D spatial information are incapable of imaging the very dynamic processes discussed here. For example, to realise data even at half the time resolution presented in this paper, the sample should be rotated at 15,000 rpm and more than 100k images would be required, imaging at 300 kHz. This is technically unfeasible for synchrotron imaging, currently, and imaging through a relatively thick and dense metal further increases the technical challenge. Even when the technical requirements related to photon flux and fast precision rotation are met, such fast rotations would distort the flow in the melt pool due to inertial and rotational forces.

Our findings provide insight into internal melt pool flow during arc melting. The widely used arc welding and the emerging arc additive manufacturing routes can be optimised by controlling internal melt flow through adjusting surface active elements.

## Methods

**Materials and welding**. 10 mm (diameter) × 50 mm (length) samples were used for the experiments. Low S and high S steels were chosen to study the effect of S. The low S steel had a nominal chemical composition (wt.%) of C—0.145, Mn—1.02, Si—0.014, Fe—98.81, P—0.001, S—0.0005, Cr—0.005, and Ni—0.005, while the high S steel's nominal chemical composition was C—0.15, Mn—2.0, Si—1.0, Fe—67.35, P—0.2, S—0.3, Cr—19.0, and Ni—10.0. These samples present real industrial materials with low and high S content. In order to investigate the surface active element role, in an ideal situation one would utilise samples from the same material with different active element contents. Doping was disregarded as it would likely lead to inhomogeneous surfactant concentration. An alternative of placing S particles on the sample surface was also disregarded as the particles would be blown away unless glued and glue would introduce further chemical uncertainty. The main difference in our sample chemistry, other than S content, are the high Cr and Ni content of the S303 stainless steel. Cr and Ni are not surface active and thus do not have an influence on the driving force of the flow. During the experiments, a three second (10 V, 125 A) tungsten inert gas (TIG) spot weld was made in the centre of the specimens. The welding process used a non-consumable tungsten electrode in DCEN polarity. The target distance from the electrode tip to the workplace was 1 mm. In reality this may have increased up to 1.2 mm for some experiments. For the images presented in this paper, the exact distance between the electrode tip and workpiece was measured in ImageJ using the pixel numbers and pixel size (13 μm per pixel). At the start of the test (before welding begins), the distance is 1.027 mm for the low S sample and 1.092 mm for the high S sample. Argon was used as the shielding gas, with an 8 l min$^{-1}$ flow rate. Owing to the metallic nature of W and Ta particles, limited lives of the tracer particles were observed during the experiments. Even though both elements melting point is above the expected weld pool temperatures, their metallic nature caused them to dissolve in the weld pool. The dissolution rate was very high for Ta particles in contrast to W particles. As such, Ta tracers were not effective for more than a fraction of a second. However, W particles survived for a few seconds to provide quantitative information of the flow velocities in the weld pools.

**Fast synchrotron radiography**. Experiments were performed at the I12 Joint Engineering, Environment and Processing beamline of Diamond Light Source, UK. Polychromatic (white) beam of ~50–150 keV delivered by a 4.2 T superconducting multipole wiggler was used to maximise the X-ray photon flux at the sample position, located approximately 50 m away from the source. The beam size was 12 × 50 mm$^2$ ($H × W$). The beam was transmitted through the entire weld pool. The detector was a Vision Research Phantom v7.3 CMOS camera, lens-coupled to cadmium tungstate and cesium iodide scintillators. With an optical magnification of ×1.8, the linear resolution was 13 μm per pixel. Imaging was acquired at frame rates up to 2 kHz at 800 × 600 pixels per frame.

**Image analysis and quantification**. All image analysis was completed using ImageJ software[60]. To retain image integrity, minimal processing was implemented to the raw data. The routine started by applying a 3D Hybrid median filter to eliminate high contrast speckle noise. In order to observe the weld geometry, minimum intensity stacks were made of all the images in a test sequence. The overlay of the tracking particles, coupled with the density increase in the liquid phase, results in a clearly visible (darker) region where the molten weld had been during welding. The increased density of the liquid phase has been attributed due to the partial melting of tantalum particles during welding. The tantalum enriched weld offers a higher absorption co-efficient to that of the plain steel. Fluid flow analysis was carried out using the manual tracking plugin to ImageJ. This plugin provides a way to retrieve $XY$ coordinates as well as velocity, distance covered between two frames and intensity of the selected pixel. In order to realise

satisfactorily precise quantification flow velocities, selection of tracers that moves relative in the plane orthogonal to the synchrotron beam was essential. To ensure selection of orthogonal in-plane flow, only particles that flowed at the extremities of the weld pool were tracked for quantifications. The flow velocities indicated here from the experiments are actually the travelling velocities of the tracer particles. As these tracer particles are heavier than the metallic melts in the pool, 100% momentum transfer cannot be anticipated and a minor discrepancy, in favour of higher flow velocities, can be expected.

## Data availability

Representative samples of the research data are given in the figures. Other datasets generated and/or analysed during this study are not publicly available due to their large size but are available from the corresponding author on reasonable request.

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

## Acknowledgements

This research work was supported by the European Commission as part of the FP7 programme, as the project, Modelling of Interface Evolution in Advanced Welding, Contract No. NMP3-SL-2009-229108. We thank Diamond Light Source for access to X-ray beamtime on Beamline I12 (visits EE8218, EE7855).

## Author contributions

L.A. took part in experiments, led the analysis and wrote the manuscript with assistance from all authors. H.B.D. instigated and supervised the project and designed experiments. W.U.M. took part in the experiments and data analysis, assisted in writing the manuscript. R.A. co-designed the experiments (beamline and imaging) and assisted in data analysis, A.K. took part in the experiments and flow analysis. S.G. took part in flow analysis, S.W.W. co-designed the experiments and took part in the experiments. J.A.M. took part in the experiments (welding). S.F. took part in data analysis and assisted in writing the manuscript. M.T. took part in the data analysis. T.C. took part in the experimental design (beamline). M.D. took part in the experimental design (beamline). C.R.K. took part in flow analysis. I.M.R. took part in the flow analysis. D.B. took part in the data analysis and experimental design. R.H.M. took part in experimental design. H.V.A. took part in experimental design and data analysis.

## Additional information

**Competing interests:** The authors declare no competing interests.

