## [Peer Review File · Nature Communications]

Reviewers' comments:

Reviewer #1 (Remarks to the Author):

1. Major claim of the paper: Using high-energy synchrotron radiation to reveal flow patterns and determine fluid velocities in weld pools.
2. Novelty: Yes, it is novel.
3. Interest: Yes, it is of high interest to the welding community. It can be of interest to the additive manufacturing community because metal additive manufacturing is also a welding process, just small in scale.
4. Convincing or not: First, yes, the observed flow patterns are very convincing to me. We developed the first model to calculate the weld pool shape by heat transfer and fluid flow. We also did visualization of flow in simulated transparent weld pools of NaNO₃. But metal weld pools are opaque, and I am impressed that the authors revealed fluid flow inside metal weld pools. Not only that, they confirmed the dramatic effect of sulfur (S) on the flow pattern, which could only be speculated so far because no one could see the flow pattern in metal weld pools. The authors showed that in a weld pool with a high S content, the flow is inward along the pool surface and downward along the pool axis, leading to a much deeper pool. This is consistent with both computer simulations and welding experiments.
Second, the measured fluid velocities seem less convincing. The authors may want to comment on the effect of the very small workpiece diameter (10 mm) on the surface temperature gradients and hence the maximum surface velocities. The diameter made it easier for x-ray to penetrate the workpiece but could have also limited the maximum temperature at the center of the pool surface that was needed to drive the flow as fast as that in real welding. A significantly higher welding current and/or longer welding time could have raised the maximum temperature but could have also melted the edge of the workpiece surface. This may explain why the measured velocities (0.5 m/s with high S and 0.3 m/s with low S) were lower than expected, e.g., 1 m/s.
Third, regarding surface turbulence, the authors should at least mention the paper "Oscillatory Marangoni Flow: A Fundamental Study by Conduction-Mode Laser Spot Welding", published by S. Kou, C. Limmaneevichitr and P. S. Wei in the *Welding Journal*, vol. 90, 2011, pp. 229-s to 240-s. It is closely related to the present manuscript, also examining a stationary weld pool. Surface oscillation was observed in the low-S (40 ppm S) weld but not in the high-S (140 ppm) weld.
Fourth, another factor to point out is the very S content of the high-S steel, i.e., 0.3 wt% S (3,000 ppm?). This S level looks more like that in steels intended for free machining than that in structural steels intended for welding. Data of how the surface-tension varies with temperature are available (from DebRoy and coworkers) at levels like 200 or 300 ppm. But I am not sure about 0.3 wt% S. Steels with a S level of about 200 or 300 ppm might be a better choice for the workpiece.
To conclude, I recommend acceptance of the manuscript for publication but encourage the authors to address the four comments made above.

Reviewer #2 (Remarks to the Author):

The manuscript presented the real-time weld pool evolution both for the steels with different active element (low sulfur steel and high sulfur stainless steel) by high energy synchrotron radiation experiments with the maximum temporal resolution, 1ms. The traces elements (W or Ta) movement process and the weld pool shape evolution can be captured precisely and shown clearly. The results confirmed the different liquid pool convection modes directly, inward convection for high sulfur steel and outward convection for low sulfur steel. The experimental method is novel, and the result is helpful to understand the role of the active element on the TIG weld pool convection and weld shape variation further. The final conclusion in the paper was also discussed and obtained by the welding process simulation for the liquid pool flow [1-9] and the theoretically analysis for the final weld pool shape by the temperature coefficient of surface tension [10-17].
Suggestion and question:

- (1) For TIG welding, the weld shape depends to a large extent both on the heat conduction and the heat convection in the welding process which related to the materials' thermal-physical properties and the weld pool flow mode. Generally, in order to investigate the surface active element role, it is better to carry out the experiments on the same material with different active element contents. In the manuscript, why select two kind steels as the welding substrate, low carbon steel with low sulfur and stainless steel with high sulfur? It is better to give some explanation on the materials' characteristic effect on the weld pool shape.
- (2) The welding parameters are same for the two experiments, set at 125A and 10V. In Fig.2, the distance from the electrode tip to welding plate surface seems different, relative large for low sulfur carbon steel, and small for high sulfur stainless steel. It is better to claim the electrode tip to work piece distance in the manuscript, which is also an important parameter influencing the arc current density and the heat density.
- (3) Fig.2 has been shown five times in the manuscript, are the images taken from different direction?

- [1] C.R.Heiple and J.R.Roper, Mechanism for minor element effect on GTA fusion zone geometry, *Weld. J.*, Vol.61, 1982, 97s;
- [2] C.R.Heiple, J.R.Roper, R.T.Stagner and R.J.Aden, Surface active element effects on the shape of GTA, laser and electron beam welds, *Weld.J.*, Vol.62, 1983, 72s;
- [3] C.R.Heiple and J.R.Roper, Effect of selenium on GTAW fusion zone geometry, *Weld.J.*, Vol.60, 1981, 143s;
- [4] M.Tanaka, H.Terasaki, M.Ushio and J.Lowke, Numerical study of a free-burning argon arc with anode melting, *Plasma Chemistry and Plasma Processing*, Vol.23, No.3, 2003, 585;
- [5] M.Tanaka, H.Terasaki, M.Ushio and J.Lowke, Numerical study of weld formations for stationary TIG arc in different gaseous atmosphere, The 56th Annual Assembly of International Institute of Welding, Bucharest, Romania, 2003, IIW Doc. 212-1040-03;
- [6] S.Leconte, P.Paillard, P.Chapelle, G.Henrion and J.Saindrenan, Effect of oxide fluxes on activation mechanisms of tungsten inert gas process, *Sci. & Tech. of Weld. & Join.*, Vol.11, No.4, 2006, 389;
- [7] P.Sahoo, T.DebRoy and M.J.McNallan, Surface tension of binary metal—surface active solute systems under conditions relevant to welding metallurgy, *Metall. Trans. B*, Vol.19, No.6, 1988, 483;
- [8] S.Kou, Weld pool convection and evaporation, *Welding Metallurgy*, A Wiley-interscience Publication, 1987, 91;
- [9] S.P.Lu, H.Fujii, H.Sugiyama and K.Nogi, Mechanism and Optimization of Oxide Fluxes for Deep Penetration in GTA Welding,, *Metall.& Mater.Trans.A*, Vol.34, 2003, 1901;
- [10] S. Kou and Y.H. Wang. Three-dimensional convection in laser melted pools. *Metallurgical Transactions A*. 1986, 17(12): 2265-2270
- [11] S. Kou and Y.H. Wang. Weld pool convection and its effect. *Welding Journal*. 1986(65): 63s-70s
- [12] S. Kou and Y.H. Wang. Computer simulation of convection in moving arc weld pools. *Metallurgical Transactions A*. 1986, 17(12): 2271-2277
- [13] T. Zacharia, S.A. David, J.M. Vitek and H.G. Krans. Computational modeling of stationary gas-tungsten-arc weld pool and comparison to stainless steel 304 experimental result. *Metallurgical Transactions B*. 1991, 22(2): 243-257
- [14] Y. Wang and H.L. Tsai. Effects of surface active elements on weld pool fluid flow and weld penetration in gas metal arc welding. *Metallurgical and Materials Transactions B*. 2001, 32(3): 501-515
- [15] Y. Wang, Q. Shi and H.L. Tsai. Modeling of the effects of surface-active elements on flow patterns and weld penetration. *Metallurgical and Materials Transactions B*. 2001, 32(1): 145-161
- [16] Y.Z. Zhao, Y.P. Lei and Y.W. Shi. Effects of surface-active elements sulfur on flow patterns of welding pool. *Journal of Materials Science & Technology*. 2005, 21(3): 408-414
- [17] R.H. Zhang and D. Fan. Numerical simulation of effects of activating flux on flow patterns and weld penetration in ATIG welding. *Science and Technology of Welding and Joining*. 2007, 12(1): 15-23

Reviewer #3 (Remarks to the Author):

The work is novel and interesting, however the reviewer has a number of questions regarding the results and outcomes of the work. Please see marked up manuscript for details. These need to be clarified before publication can be recommended in such a highly regarded Journal/

[Editorial Note: In their review of the first version of this manuscript, reviewer #3 added their comments to the manuscript file. These comments, excluding minor textual revisions, have been copied into this Peer Review File.

Comments on Manuscript:

I.94, 'Tungsten (W) and tantalum..': did these not affect the flow and resultant structure?

Fig 2, first instance: images should be enlarged for clarity

Fig 2: this figure is inserted in error by poor word formatting

I.123, 'yet the heat..': but they have different melt temps so different amount of energy inputs would be required to achieve a given weld pool depth and width. this result doesn't seem unexpected.

Fig 2, second instance: split these figs up and make them bigger

Fig 2: size? width or depth? looks like depth...wouldnt area be better measure?

Fig 2, third instance: pdf conversion gone very wrong

Fig 2, fifth instance: how repeatable are these trends on the same material? could do tests ex situ and compare variation in fusion zone sizes for many samples

I.203, 'though..': I am not convinced...would need to do tests with different heat inputs relative to melting temp

I.245, 'met each..': would be good to comment on how this information can be used

I.246: no conclusions section?

Point by point response to reviewer comments

The authors welcome the comments raised by the reviewers. A point-by-point response to the specific points raised is detailed in the red text below. The authors have amended the manuscript accordingly with the changes highlighted by red text in the marked up manuscript. The amended manuscript has then been proof read several times by the group of authors.

Reviewer #1

Comment 1.1:

Major claim of the paper: Using high-energy synchrotron radiation to reveal flow patterns and determine fluid velocities in weld pools.

We agree, no action required.

Comment 1.2:

Novelty: Yes, it is novel.

We agree, no action required.

Comment 1.3:

Interest: Yes, it is of high interest to the welding community. It can be of interest to the additive manufacturing community because metal additive manufacturing is also a welding process, just small in scale.

We agree, no action required.

Comment 1.4:

Convincing or not: First, yes, the observed flow patterns are very convincing to me. We developed the first model to calculate the weld pool shape by heat transfer and fluid flow. We also did visualization of flow in simulated transparent weld pools of NaNO₃. But metal weld pools are opaque, and I am impressed that the authors revealed fluid flow inside metal weld pools. Not only that, they confirmed the dramatic effect of sulfur (S) on the flow pattern, which could only be speculated so far because no one could see the flow pattern in metal weld pools. The authors showed that in a weld pool with a high S content, the flow is inward along the pool surface and downward along the pool axis, leading to a much deeper pool. This is consistent with both computer simulations and welding experiments.

We agree, no action required.

It is very humbling to receive such complimentary feedback from such highly regarded peers whose research set the foundations for the work which we have presented in this paper.

Comment 1.5:

Second, the measured fluid velocities seem less convincing. The authors may want to comment on the effect of the very small workpiece diameter (10 mm) on the surface temperature gradients and hence the maximum surface velocities. The diameter made it easier for x-ray to penetrate the workpiece but could have also limited the maximum temperature at the center of the pool surface that was needed to drive the flow as fast as that in real welding. A significantly higher welding current and/or longer welding time could have raised the maximum temperature but could have also melted the edge of the workpiece surface. This may explain why the measured velocities (0.5 m/s with high S and 0.3 m/s with low S) were lower than expected, e.g., 1 m/s.

Response

We agree that the effect of the small workpiece diameter on the surface temperature gradients and hence the maximum surface velocities should be discussed. As the reviewer alludes, the 10 mm sample thickness wasn't selected by chance, it was specifically chosen as it allows for good penetration of the X-ray spectrum which provides for good image quality with short exposure time (high frame rates). Therefore, the selected sample sizes provide high resolution images for subsequent analysis.

Assuming the material doesn't melt, we can approximate how long it takes for the energy from the arc to reach the boundaries of our work piece. We can introduce a diffusion time scale as $\tau = \frac{L^2}{\alpha}$, where α is the thermal diffusivity of the base material, and L is a length scale, in our case 5mm.

Material	Thermal conductivity [W/mK]	Density [kg/m ³]	Heat capacity [J/kgK]	Thermal diffusivity [m ² /s]	Time scale [s]
High-S steel	16	8000	600	3.3*10 ⁻⁶	7.5
Low-S steel	50	7200	800	8.7*10 ⁻⁶	2.9

As you can see in the table, for the High-S steel, it takes more than 7s for the temperature to be felt at the boundary of the work piece, for Low-S steel 2.7s. In these experiments, the arc was ignited for 3.0s. In reality, the time will be further delayed since some of the energy put in will be consumed in the phase change process.

We, therefore, conclude the work piece dimensions do not significantly influence our observations.

The reviewer's comment does hold for other reasons:

1. We are essentially measuring the speed in a 2D projection, which may be much lower than the true speed in three dimensions, as we will likely never encounter a particle travelling exactly perpendicular to the imaging plane.
2. The largest flow velocities are at the surface of the weld pool, since this is where the driving forces act. We only measure the bulk flow velocities, which are expected to be lower. If we had a camera observing the surface from above, we very well may have measured velocities up to 2m/s. See e.g. the papers referenced in the response to the comment below.
3. Referring to computed results in literature may be misleading, since most papers neglect turbulence in their computations. However, turbulence will reduce the maximum velocity obtained due to increased diffusivity (both thermal and momentum).

Page 6, paragraph 4, the following text have been added to address the above.

“The flow velocities quantified in this study must be assessed in relation to the melt pool in which they exist, the measured velocity is lower than those predicted value of 1 m s^{-1} by modelling in very large size melt pool. In this study, the 10 mm sample was specifically chosen as it was the maximum size feasible for the X-ray beam to penetrate and provide high resolution images for subsequent analysis. Other factors may also affect the discrepancy between the measured velocity and the simulated ones, including: (1) we are essentially measuring the speed in a 2D projection, which may be much lower than the true speed in three dimensions, as we will likely never encounter a particle travelling exactly perpendicular to the imaging plane, (2) referring to computed results in literature may be misleading, since most papers neglect turbulence in their computations. However, turbulence will reduce the maximum velocity obtained due to increased thermal and momentum diffusivities.”

Comment 1.6:

Third, regarding surface turbulence, the authors should at least mention the paper “Oscillatory Marangoni Flow: A Fundamental Study by Conduction-Mode Laser Spot Welding, published by S. Kou, C. Limmaneevichitr and P. S. Wei in the Welding Journal, vol. 90, 2011, pp. 229-s to 240-s. It is closely related to the present manuscript, also examining a stationary weld pool. Surface oscillation was observed in the low-S (40 ppm S) weld but not in the high-S (140 ppm) weld.

We agree that this is a very appropriate paper to cite in this instance. The bulk turbulence results presented in this research actually show the opposite trend to that observed in the paper which is recommended by the reviewer. In this study, we only observe surface turbulence in high S steel due to the competing forces within the melt pool. See images below of surface oscillations I high S steel.

Our results correspond to the findings of Kidess [18, 53], Xiao & Den Ouden [54] and Mills et al [55] who observe oscillations for steels with 180 and 530ppm of S, but not for steel with low S. In actual fact, the most impressive radiographs we have of surface oscillations are from the molten behaviour of MIG melt pools. These results are outside of the scope of this paper and are something we aim to work on for future publication.

To acknowledge the reference raised by the reviewer, the following text (and references) have been added to the discussion section, page 7, paragraph 1:

These findings are in agreement with the work of Kidess [18,53] and Xiao & Den Ouden [54] but opposite to those of Kou et al. [55].

[18] Kidess, A., Kenjereš, S. & Kleijn, C. R. The influence of surfactants on thermocapillary flow instabilities in low Prandtl melting pools. *Physics of Fluids (1994-present)* **28**, 062106+ (2016). URL <http://dx.doi.org/10.1063/1.4953797>.

[53] Kidess, A., Kenjeres, S., Righolt, B. W. & Kleijn, C. R. Marangoni driven turbulence in high energy surface melting processes. *International Journal of Thermal Sciences* **104**, 412-422 (2016). URL <http://dx.doi.org/10.1016/j.ijthermalsci.2016.01.015>.

[54] Xiao, Y. H. & Den Ouden, G. A study of GTA weld pool oscillation. *Welding Journal* **69**, S289-S293 (1990).

[55] Mills, K. C., Keene, B. J., Brooks, R. F. & Shirali, A. Marangoni effects in welding. *Mathematical, Physical and Engineering Sciences* **356**, 911-925 (1998). URL <http://dx.doi.org/10.1098/rsta.1998.0196>.

[56] Oscillatory Marangoni Flow: A Fundamental Study by Conduction-Mode Laser Spot Welding, published by S. Kou, C. Limmaneevichitr and P. S. Wei in the *Welding Journal*, vol. 90, 2011, pp. 229-s to 240-s

Comment 1.7:

Fourth, another factor to point out is the very S content of the high-S steel, i.e., 0.3 wt% S (3,000 ppm?). This S level looks more like that in steels intended for free machining than that in structural steels intended for welding. Data of how the surface-tension varies with temperature are available (from DebRoy and coworkers) at levels like 200 or 300 ppm. But I am not sure about 0.3 wt% S. Steels with a S level of about 200 or 300 ppm might be a better choice for the workpiece.

The reviewer is correct in that there is little measured data beyond 300ppm S. If we do assume the Sahoo-model to be valid at any surfactant concentration, the resulting surface tension temperature coefficient is positive up to roughly 2600 K, which is about the temperature where iron starts vaporizing, that is the coefficient will not turn negative any more (or just barely). The data reviewed by Keene [*] up to extremely high concentrations of 50%-S does not propose any conflicting mechanisms. Therefore no changes are made in the MS.

[*]Keene, B. J. Review of data for the surface tension of iron and its binary alloys. *International Materials Reviews* 1-37 (1988). URL <http://dx.doi.org/10.1179/095066088790324139>.

Comment 1.8:

To conclude, I recommend acceptance of the manuscript for publication but encourage the authors to address the four comments made above.

We very much appreciate this recommendation and the valuable points you have raised. The changes we have made to the manuscript, based on the comments you have raised, have certainly strengthened the paper.

Reviewer #2

Comment 2.1:

The manuscript presented the real-time weld pool evolution both for the steels with different active element (low sulfur steel and high sulfur stainless steel) by high energy synchrotron radiation experiments with the maximum temporal resolution, 1ms. The traces elements (W or Ta) movement process and the weld pool shape evolution can be captured precisely and shown clearly. The results confirmed the different liquid pool convection modes directly, inward convection for high sulfur steel and outward convection for low sulfur steel. The experimental method is novel, and the result is helpful to understand the role of the active element on the TIG weld pool convection and weld shape variation further. The final conclusion in the paper was also discussed and obtained by the welding process simulation for the liquid pool flow **[1-9]** and the theoretical analysis for the final weld pool shape by the temperature coefficient of surface tension **[10-17]** .

[1] C.R.Heiple and J.R.Roper, Mechanism for minor element effect on GTA fusion zone geometry, Weld. J., Vol.61, 1982, 97s;

[2] C.R.Heiple, J.R.Roper, R.T.Stagner and R.J.Aden, Surface active element effects on the shape of GTA, laser and electron beam welds, Weld.J., Vol.62, 1983, 72s;

[3] C.R.Heiple and J.R.Roper, Effect of selenium on GTAW fusion zone geometry, Weld.J., Vol.60, 1981, 143s;

[4] M.Tanaka, H.Terasaki, M.Ushio and J.Lowke, Numerical study of a free-burning argon arc with anode melting, Plasma Chemistry and Plasma Processing, Vol.23, No.3, 2003, 585;

[5] M.Tanaka, H.Terasaki, M.Ushio and J.Lowke, Numerical study of weld formations for stationary TIG arc in different gaseous atmosphere, The 56th Annual Assembly of International Institute of Welding, Bucharest, Romania, 2003, IIW Doc. 212-1040-03;

[6] S.Leconte, P.Paillard, P.Chapelle, G.Henrion and J.Saindrean, Effect of oxide fluxes on activation mechanisms of tungsten inert gas process, Sci. & Tech. of Weld. & Join., Vol.11, No.4, 2006, 389;

[7]. P.Sahoo, T.DebRoy and M.J.McNallan, Surface tension of binary metal—surface active solute systems under conditions relevant to welding metallurgy, Metall. Trans. B, Vol.19, No.6, 1988, 483;

[8] S.Kou, Weld pool convection and evaporation, Welding Metallurgy, A Wiley-interscience Publication, 1987, 91;

[9] S.P.Lu, H.Fujii,H.Sugiyama and K.Nogi, Mechanism and Optimization of Oxide Fluxes for Deep Penetration in GTA Welding,, Metall.& Mater.Trans.A, Vol.34, 2003, 1901;

[10] S. Kou and Y.H. Wang. Three-dimensional convection in laser melted pools. Metallurgical Transactions A. 1986, 17(12): 2265-2270

[11] S. Kou and Y.H. Wang. Weld pool convection and its effect. Welding Journal. 1986(65): 63s-70s

[12] S. Kou and Y.H. Wang. Computer simulation of convection in moving arc weld pools. Metallurgical Transactions A. 1986, 17(12): 2271-2277

[13] T. Zacharia, S.A. David, J.M. Vitek and H.G. Krans. Computational modeling of stationary gas-tungsten-arc weld pool and comparison to stainless steel 304 experimental result. Metallurgical Transactions B. 1991, 22(2): 243-257

[14] Y. Wang and H.L. Tsai. Effects of surface active elements on weld pool fluid flow and weld penetration in gas metal arc welding. Metallurgical and Materials Transactions B. 2001, 32(3): 501-515

[15] Y. Wang, Q. Shi and H.L. Tsai. Modeling of the effects of surface-active elements on flow patterns and weld penetration. Metallurgical and Materials Transactions B. 2001, 32(1): 145-161

[16] Y.Z. Zhao, Y.P. Lei and Y.W. Shi. Effects of surface-active elements sulfur on flow patterns of welding pool. Journal of Materials Science & Technology. 2005, 21(3): 408-414

[17] R.H. Zhang and D. Fan. Numerical simulation of effects of activating flux on flow patterns and weld penetration in ATIG welding. Science and Technology of Welding and Joining. 2007, 12(1): 15-23

We agree, many of the references which the reviewer has listed are already included within the original manuscript.

For completeness, those not included [3-10, 13, 15-16] have been added to the re-submission, where appropriate, to fully credit previous work and also increase the visibility of the current research's impact.

Comment 2.2:

For TIG welding, the weld shape depends to a large extent both on the heat conduction and the heat convection in the welding process which related to the materials' thermal-physical properties and the weld pool flow mode. Generally, in order to investigate the surface active element role, it is better to carry out the experiments on the same material with different active element contents. In the manuscript, why select two kind steels as the welding substrate, low carbon steel with low sulfur and stainless steel with high sulfur? It is better to give some explanation on the materials' characteristic effect on the weld pool shape.

These samples present real industrial materials with low and high S content. We didn't have the possibility to create our own alloys, so we had to resort to commercially available alloys. We could have done tests with the same alloy, doping the surface with S, but a) we saw problems with our tungsten particles being blown away unless "glued", which introduces other substances which may react, and b) we would obtain a very unpredictable, possibly inhomogeneous surfactant concentration. The main difference between the chemistries of the substrates, other than S content, are the high Chrome and Ni content of the S303 stainless steel with high S content. Chrome and Nickel are not surface active and thus do not have an influence on the driving force. They do have some influence on the thermodynamic properties of the material, namely the thermal conductivity, but this does not affect the main message of this paper in our opinion.

Following text have been added to address the above in page 8, Materials and welding

"These samples present real industrial materials with low and high S content. In order to investigate the surface active element role, in an ideal situation would utilise, samples from the same material with different active element contents. Doping was disregarded as it would likely lead to

inhomogeneous surfactant concentration. An alternative of placing S particles on the sample surface was also disregarded as the particles would be blown away unless “glued” and glue would introduce further chemical uncertainty. The main difference in our sample chemistry, other than S content, are the high Chrome and Ni content of the S303 stainless steel. Chrome and Nickel are not surface active and thus do not have an influence on the driving force of the flow.”

Comment 2.3:

The welding parameters are same for the two experiments, set at 125A and 10V. In Fig.2, the distance from the electrode tip to welding plate surface seems different, relative large for low sulfur carbon steel, and small for high sulfur stainless steel. It is better to claim the electrode tip to work piece distance in the manuscript, which is also an important parameter influencing the arc current density and the heat density.

The target distance from the electrode tip to the workplace was 1 mm. In reality this may have increased up to 1.2 mm for some experiments. However, for analysis and publication within this paper, we have intentionally chosen samples with the most representative electrode distances. For the images we present in this paper, we can measure the exact distance between the electrode tip and workpiece in ImageJ using the pixel numbers and pixel size (13 $\mu\text{m}/\text{pixel}$). At the start of the test (before welding begins), the distance is 1.027 mm for the low S sample and 1.092 mm for the high S sample. So the difference is negligible. The distance may seem less for the high-S sample once welding begins as the liquid surface bulges upwards due to the emerging flow pattern, and this is beyond our means of control.

To clarify this, the following text has been added to the methods section, fast synchrotron radiography heading

“The target distance from the electrode tip to the workplace was 1 mm. In reality this may have increased up to 1.2 mm for some experiments. For the images presented in this paper, the exact distance between the electrode tip and workpiece was measured in ImageJ using the pixel numbers and pixel size (13 $\mu\text{m}/\text{pixel}$). At the start of the test (before welding begins), the distance is 1.027 mm for the low S sample and 1.092 mm for the high S sample.”

Comment 2.4:

Fig.2 has been shown five times in the manuscript, are the images taken from different direction?

No, the images are identical. There was an error in the PDF conversion when uploading the original Word.docx file. The conversion error has replicated Figure 2 several times throughout the online submission file. This will be rectified and checked in the re-submission.

Reviewer #3:

Comment 3.1:

The work is novel and interesting, however the reviewer has a number of questions regarding the results and outcomes of the work. Please see marked up manuscript for details. These need to be clarified before publication can be recommended in such a highly regarded Journal.

Agreed. The points highlighted in the marked up manuscript are addressed in detail in the forthcoming text.

Comment 3.2:

Line 94 of reviewer #3s marked up manuscript - did these not affect the flow and resultant structure?

It is envisaged that the particles would have an effect on the final microstructure as we are essentially adding Ta and W based inclusions to the fusion zone. However, in the current contribution we are not particularly concerned with the final microstructure. The main focus of this study is the liquid stage of welding and in this instance the particles have negligible effect on the resultant flow patterns and velocities as the sinking speed is negligible in comparison to the very dynamic flow velocities within the melt pool. Certainly the influence of S levels in reversing Marangoni flow direction has not been masked by the presence of these particles.

To clarify this point, the following text has been added to page 4, paragraph 2:

“While the particles will likely affect the microstructure of the solidified fusion zone, they have an insignificant effect on the flow patterns and velocities in the liquid melt pool as the sinking speed is negligible in comparison to the highly dynamic flow velocities characterised.”

Comment 3.3:

Line 110 of reviewer #3s marked up manuscript - images should be enlarged for clarity

We disagree with this, but would welcome contrary views! We think the figure presents the data clearly and in a succinct single figure.

Comment 3.4:

Line 110 of reviewer #3s marked up manuscript - this figure is inserted in error by poor word formatting

We agree. There was an error in the PDF conversion when uploading the original Word.docx file. The conversion error has replicated Figure 2 several times throughout the online submission file. This will be rectified and checked in the re-submission.

Comment 3.5:

Line 123 of reviewer #3s marked up manuscript - but they have different melt temps so different amount of energy inputs would be required to achieve a given weld pool depth and width. This result doesn't seem unexpected.

The latent heat of the alloys would have a much greater effect on the energy needed to melt them than would the liquidus (melting) temperature. The low S alloy is predominantly Fe (nearly 99 wt.%), but the high S alloy has 19wt.% Cr and 10wt% Ni: both of these alloying elements have higher latent heats than Fe, so a rule-of-mixtures approach suggest the high S alloy has a higher latent heat of fusion than the low S alloy. Yet the volume melted for the high S alloy is actually larger: this means that the phenomena we suggest in the paper, due primarily to surface-tension-driven convection, could even be stronger (and definitely not weaker) than we claim in the manuscript.

(We would have this as a point in the response to reviewers, and not make a change or addition to the MS itself).

Comment 3.6:

Line 125 of reviewer #3s marked up manuscript - split these figs up and make them bigger.

There was an error in the PDF conversion when uploading the original Word.docx file. The conversion error has replicated Figure 2 several times throughout the online submission file. This will be rectified and checked in the re-submission.

Comment 3.7:

Line 125 of reviewer #3s marked up manuscript - size? width or depth? looks like depth...wouldn't area be better measure?

Apologies, but we think that this is quite clearly described within Figure 2 in its current/original state. Figure 2(m) shows the melt pool size evolution in terms of both depth and width. This is clearly denoted by the figure legend label below the plot. Assuming the weld is a hemispherical shape, the area can be estimated using the depth and width. The area is estimated and discussed within the main text. The total volume of alloy melted after 2000 ms is approx. 200% greater in the high S (~72.47 mm³) sample compared to the low S (~23.84 mm³) sample, yet the heat input to both alloys is the same.

The markers within Figure 2(m) and (n) have been changed in the revised manuscript to address the point raised by the reviewer and further distinguish the datasets plotted within Figure 2.

Comment 3.8:

Line 136 of reviewer #3s marked up manuscript - pdf conversion gone very wrong.

We agree. There was an error in the PDF conversion when uploading the original Word.docx file. The conversion error has replicated Figure 2 several times throughout the online submission file. This will be rectified and checked in the re-submission.

Comment 3.9:

Line 149 of reviewer #3s marked up manuscript - how repeatable are these trends on the same material? could do tests ex situ and compare variation in fusion zone sizes for many samples?

These trends are very repeatable. The subject of melt pool fluid flow and geometrical evolution has been the focus of significant research over many years. Ex situ studies, to which the reviewer alludes, have already been conducted in several of the studies referenced within this paper and confirm the repeatability of the trends we report in our study.

Comment 3.10:

Line 203 of reviewer #3s marked up manuscript - I am not convinced...would need to do tests with different heat inputs relative to melting temp

This is a well-known fact (as shown in Ref 49, 50 of revised manuscript) on the risen and shallow shrunken centres of high and low S steels irrespective of weld parameters. In addition, reviewer's point on melting temperatures are explained using the concept of the latent heat. Our data in figure 2 shows that there is a real qualitative difference beyond a time/energy delay. Therefore no changes are made in MS.

Comment 3.11:

Line 245 of reviewer #3s marked up manuscript - would be good to comment on how this information can be used

We agree, the following text has been added to the end of the discussion section to address this comment:

“Our findings provide insight into internal melt pool flow during arc melting. The widely used arc welding and the emerging arc additive manufacturing routes can be optimised by controlling internal melt flow through adjusting surface active elements.”

Comment 3.12:

Line 245 of reviewer #3s marked up manuscript - no conclusions section?

Indeed, there is no conclusions section. Nature Communications formatting guidelines specify that there should be no conclusions section. No action required.

Typographical errors:

Line 42 of reviewer #3s marked up manuscript – “as” changed to “as a” as suggested

Line 228 of reviewer #3s marked up manuscript – “dependant” changed to “dependent” as suggested

In Addition

We have revised our abstract according to the Guidelines from Nature Comms:

ABSTRACT		
No more than 150 words		
Does not contain references		
Results of the current study are written in present tense		
Starts with short description of background (2-3 sentences)		
Continues with presentation of the major results ('Here we show' or similar)		
Ends with a description of the paper's conclusion		

The new abstract contains 144 words and provide the required info and is present in the right order as required above:

Internal flow behaviour during melt-pool-based metal manufacturing remains unclear and hinders progression to manufacturing process improvement. We present direct time-resolved imaging of melt pool flow dynamics in a novel high-energy synchrotron radiation experiment. We track internal flow streams for the first time during arc welding of steel and measure instantaneous flow velocities ranging from 0.1 m/s to 0.5 m/s. When the temperature-dependent surface tension coefficient is negative, bulk turbulence is the main flow mechanism and the critical velocity for surface turbulence is below the limits identified in previous theoretical studies. When the alloy exhibits a positive temperature-dependent surface tension coefficient, surface turbulence occurs and derisory oxides can be entrapped within the subsequent solid as result of higher flow velocities. The widely used arc welding and the emerging arc additive manufacturing routes can be optimised by controlling internal melt flow through adjusting surface active elements.

REVIEWERS' COMMENTS:

Reviewer #1 (Remarks to the Author):

The authors have addressed all of my comments well. I recommend acceptance of the manuscript for publication.

Reviewer #2 (Remarks to the Author):

The welding method applied in the paper is gas tungsten arc welding (self-fusible GTAW or TIG), and there is no additive consumable material applied to fill in the weld pool. For the other fusion welding methods with filler metal, such as MAG, MIG, SAW, the internal flow behavior is more complex. Therefore, I suggest the title of the manuscript should be changed to "Revealing internal flow behavior in gas tungsten arc welding".

Reviewer #3 (Remarks to the Author):

The response to reviewers comments have been well addressed. I recommend the paper is now published

Point by point response to reviewer comments

The authors are delighted that the manuscript has been accepted in principle. A point-by-point response to the final reviewer comments is detailed in the red text below. The authors have amended the manuscript accordingly with the changes tracked using Microsoft Word tracked changes tool. The amended manuscript has then been proof read several times by the group of authors.

REVIEWERS' COMMENTS:

Reviewer #1 (Remarks to the Author):

The authors have addressed all of my comments well. I recommend acceptance of the manuscript for publication.

Thankyou. No action required.

Reviewer #2 (Remarks to the Author):

The welding method applied in the paper is gas tungsten arc welding (self-fusible GTAW or TIG), and there is no additive consumable material applied to fill in the weld pool. For the other fusion welding methods with filler metal, such as MAG, MIG, SAW, the internal flow behavior is more complex. Therefore, I suggest the title of the manuscript should be changed to "Revealing internal flow behavior in gas tungsten arc welding".

We accept that in additive processes continues addition of materials add further complexity to the process. However, we believe the difference in flow patterns to be restricted to the pool surface and its oscillation, while the internal bulk flow, which we have been visualized in our experiments, will remain essentially the same. In our manuscript we reveal the basic governing conditions for metallic melt pools, which are the base for arc additive manufacturing processes in addition to the arc welding. We believe our manuscript provide essential basic scientific information related to arc additive manufacturing processes. Thus, adhere to our original title facilitate to receive due attention and readability from the emerging additive manufacturing scientific community, who can be vastly benefited from our work. We want to use our original title.

Reviewer #3 (Remarks to the Author):

The response to reviewers comments have been well addressed. I recommend the paper is now published

Thankyou. No action required.